# Is $L^2$ Physics-Informed Loss Always Suitable for Training Physics-Informed Neural Network?

**Chuwei Wang**[1*], **Shanda Li**[2,5*], **Di He**[3†], **Liwei Wang**[3,4†]

[1]School of Mathematical Sciences, Peking University
[2]Machine Learning Department, School of Computer Science, Carnegie Mellon University
[3]National Key Laboratory of General Artificial Intelligence,
School of Intelligence Science and Technology, Peking University
[4] Center for Data Science, Peking University   [5] Zhejiang Lab
chuwei.wang@pku.edu.cn, shandal@cs.cmu.edu
dihe@pku.edu.cn, wanglw@pku.edu.cn

## Abstract

The Physics-Informed Neural Network (PINN) approach is a new and promising way to solve partial differential equations using deep learning. The $L^2$ Physics-Informed Loss is the de-facto standard in training Physics-Informed Neural Networks. In this paper, we challenge this common practice by investigating the relationship between the loss function and the approximation quality of the learned solution. In particular, we leverage the concept of stability in the literature of partial differential equation to study the asymptotic behavior of the learned solution as the loss approaches zero. With this concept, we study an important class of high-dimensional non-linear PDEs in optimal control, the Hamilton-Jacobi-Bellman (HJB) Equation, and prove that for general $L^p$ Physics-Informed Loss, a wide class of HJB equation is stable only if $p$ is sufficiently large. Therefore, the commonly used $L^2$ loss is not suitable for training PINN on those equations, while $L^\infty$ loss is a better choice. Based on the theoretical insight, we develop a novel PINN training algorithm to minimize the $L^\infty$ loss for HJB equations which is in a similar spirit to adversarial training. The effectiveness of the proposed algorithm is empirically demonstrated through experiments. Our code is released at `https://github.com/LithiumDA/L_inf-PINN`.

## 1   Introduction

Recently, with the explosive growth of available data and computational resources, there have been growing interests in developing machine learning approaches to solve partial differential equations (PDEs) [14, 13, 33, 28]. One seminal work in this direction is the Physics-Informed Neural Network (PINN) approach [28] which parameterizes the PDE's solution as a neural network. By defining differentiable loss functionals that measure how well the model fits the PDE and boundary conditions, the network parameters can be efficiently optimized using gradient-based approaches. $L^2$ distance is one of the most popularly used measures, which calculates the $L^2$ norm of the PDE and boundary residual on the domain and boundary, respectively. Previous works demonstrated that PINN could solve a wide range of PDE problems using the $L^2$ Physics-Informed Loss, such as Poisson equation, Burgers' equation, and Navier-Stokes equation [28, 6].

Although previous works empirically demonstrated promising results using $L^2$ Physics-Informed Loss, we argue the plausibility of using this loss for (high-dimensional) non-linear PDE problems.

---

*Equal contribution.

†Correspondence to: Liwei Wang <wanglw@pku.edu.cn> and Di He <dihe@pku.edu.cn>.

36th Conference on Neural Information Processing Systems (NeurIPS 2022).

We know the trivial fact that the learned solution will equal the exact solution when its $L^2$ loss equals zero. However, the quality of a learned solution with a small but non-zero loss, which is a more realistic scenario in practice, remains unknown to have any approximation guarantees. In this work, we aim at answering a fundamental question:

*Can we guarantee that a learned solution with a small Physics-Informed Loss always corresponds to a good approximator of the exact solution?*

To thoroughly investigate the problem, we advocate analyzing the stability of PDE [8] in the PINN framework. Stability characterizes the asymptotic behavior of the distance between the learned solution and the exact solution when the Physics-Informed Loss approaches zero. If the PDE is not stable with respect to certain loss functions, we may not obtain good approximate solutions by minimizing the loss. To show the strength of the theory, we perform a comprehensive study on the stability of an important class of high-dimensional non-linear PDEs in optimal control, the Hamilton-Jacobi-Bellman (HJB) equation, which establishes a necessary and sufficient condition for a control's optimality with regard to the cost function. Interestingly, we prove that for general $L^p$ Physics-Informed Loss, the HJB equation is stable only if $p$ is sufficiently large. This finding suggests that the most widely used $L^2$ loss may not be suitable for training PINN on HJB equations as the learned solution can be arbitrarily distant from the exact solution. Empirical observation verifies the theoretical results.

We further show our theory can serve as a principled way to design loss functions for training PINN. For the high-dimensional HJB equation we target in the paper, the theoretical result suggests that $L^\infty$ loss may be a better choice to learn approximate solutions. Motivated by this insight, we propose a new algorithm for training PINN, which adopts a min-max optimization procedure to minimize the $L^\infty$ loss. Our approach resembles the well-known adversarial training framework. In each iteration, we first fix the network parameters and learn adversarial data points to approximate $L^\infty$ loss, and then optimize the network parameters to minimize the loss. When the training finishes, the learned network will converge to a solution with small $L^\infty$ losses and is close to the exact solution. We conduct experiments to demonstrate the effectiveness of the proposed algorithm. All empirical results show that our method can indeed learn accurate solutions for HJB equations and is much better than several baseline methods.

The contribution of the paper is summarized as follows.

- We make the first step towards theoretically studying the loss design in PINN, and formally introduce the concept of stability in the literature of PDE to characterize the quality of a learned solution with small but non-zero Physics-Informed Loss.

- We provide rigorous investigations on an important class of high-dimensional non-linear PDEs in optimal control, the HJB equation. Our results suggest that the widely used $L^2$ loss is not a suitable choice for training PINN on HJB equations.

- Based on the theoretical insight, we develop a novel PINN training algorithm to minimize the $L^\infty$ loss for HJB equations. We empirically demonstrate that the proposed algorithm can significant improve the accuracy of PINN in solving the optimal control problems.

## 2 Related Works

Physics-Informed Neural Network approaches [33, 28] learn to find parametric solutions to satisfy equations and boundary conditions with gradient descent. There has been a notable scarcity of papers that rigorously justify why PINNs work. Important works include [31], which prove the convergence of PINN for second-order elliptic and parabolic equations. In [21], the authors study the statistical limit of learning a PDE solution from sampled observations for elliptic equations. In [29], the convergence of PINN with $L^2$ loss is established for Navier-Stokes equations. At the same time, several works observed different failure modes for training PINN in other PDE problems. In [17], researchers discover that PINN sometimes fails to learn accurate solutions to a class of convection and reaction equations. Their analysis shows that this may be attributed to the complicated loss landscape. [34] observed PINN failed to learn the Helmholtz equation due to the incommensurability between PDE and boundary losses.

In this work, we mainly experiment with the Hamilton-Jacobi-Bellman (HJB) equation in optimal control. Previously, there were several works aiming at solving the HJB equation using deep learning methods [12, 13, 25, 39, 24, 1, 27, 5]. [13] is among the first to leverage neural networks to solve HJB equations. In particular, [13] targets constructing an approximation to a solution value $u$ at $T = 0$, which is further transformed into a backward stochastic differential equation and learned by neural networks. The main difference between [13] and ours is that [13] only learns the solution on a pre-defined time frame, while with our method, the obtained solution can be evaluated for any time frame. Recently, [12] tackled the offline reinforcement learning problem and developed a soft relaxation of the classical HJB equation, which can be learned using offline behavior data. The main difference between [12] and ours is that no additional data is required in our setting.

Stability is one of the most fundamental concepts in studying the well-posedness of PDE problems. Formally speaking, it characterizes the behavior of the solution to a PDE problem when a small perturbation modifies the operator, initial condition, boundary condition, or force term. We say the equation is stable if the solution of the perturbed PDE converges to the exact solution as the perturbations approach zero [8]. The problem regarding whether a PDE is stable has been intensively studied [8, 19, 11] in literature. There are also some works studying how (regularity) conditions affect stability. [20] and [7] investigate in which topology Couette Flow is asymptotic stable. [4] answers in which Sobolev space defocusing nonlinear Schrodinger equation, real Korteweg-de Vries (KdV), and modified KdV are locally well-posed. Our main focus is akin to the latter works but is settled in the machine learning framework.

## 3   Preliminary

In this section, we introduce basic background on Physics-Informed Neural Networks and stochastic optimal control problems. Without loss of generality, we formulate any partial differential equation as:

$$\begin{cases} \mathcal{L}u(x) = \varphi(x) & x \in \Omega \subset \mathbb{R}^n \\ \mathcal{B}u(x) = \psi(x) & x \in \partial\Omega, \end{cases} \tag{1}$$

where $\mathcal{L}$ is the partial differential operator and $\mathcal{B}$ is the boundary condition. We use $x$ to denote the spatiotemporal-dependent variable, and use $\Omega$ and $\partial\Omega$ to denote the domain and boundary.

**Physics-Informed Neural Networks (PINN)**   PINN [28] is a popular choice to learn the function $u(x)$ automatically by minimizing the loss function induced by the PDE (1). To be concrete, given $p \in (1, +\infty)$, we define the $L^p$ Physics-Informed Loss as

$$\ell_{\Omega,p}(u) = \|\mathcal{L}u(x) - \varphi(x)\|_{L^p(\Omega)}^p, \tag{2}$$

$$\ell_{\partial\Omega,p}(u) = \|\mathcal{B}u(x) - \psi(x)\|_{L^p(\partial\Omega)}^p. \tag{3}$$

The loss term $\ell_{\Omega,p}(u)$ in Eq. (2) corresponds to the PDE residual, which evaluates how $u(x)$ fits the partial differential equation on $\Omega$; and $\ell_{\partial\Omega,p}(u)$ in Eq. (3) corresponds to the boundary residual, which measures how well $u(x)$ satisfies the boundary condition on $\partial\Omega$. $L^p$ denotes $p$-norm, where $p$ is usually set to 2, leading to a "mean squared error" interpretation of the loss function [33, 28]. The goal is to find $u^*$ that minimizes a linear combination of the two losses defined above. The function $u(x)$ is usually parameterized by neural network $u_\theta(x)$ with parameter $\theta \in \Theta$. To find $\theta^*$ efficiently, PINN approaches use gradient-based optimization methods. Note that computing the loss involves integrals over $\Omega$ and $\partial\Omega$. Thus, Monte Carlo methods are commonly used to approximate $\ell_{\Omega,p}(u)$ and $\ell_{\partial\Omega,p}(u)$ in practice.

**Stochastic Optimal Control**   Stochastic control [9, 3] is an important sub-field in optimal control theory. In stochastic control, the state function $\{X_t\}_{0 \le t \le T}$ is a stochastic process, where $T$ is the time horizon of the control problem. The evolution of the state function is governed by the following stochastic differential equation:

$$\begin{cases} \mathrm{d}X_s = m(s, X_s)\mathrm{d}s + \sigma\mathrm{d}W_s & s \in [t, T] \\ X_t = x \end{cases}, \tag{4}$$

where $m : [t, T] \times \mathbb{R}^n \to \mathbb{R}^n$ is the control function and $\{W_s\}$ is a standard $n$-dimensional Brownian motion.

Given a control function $m$, its total cost is defined as $J_{x,t}(m) = \mathbb{E} \int_t^T r(X_s, m, s) \mathrm{d}s + g(X_T)$, where $r : \mathbb{R}^n \times \mathbb{R}^n \times [0, T] \to \mathbb{R}$ measures the cost rate during the process and $g : \mathbb{R}^n \to \mathbb{R}$ measures the final cost at the terminal state. The expectation is taken over the randomness of the trajectories.

We are interested in finding a control function that minimizes the total cost for a given initial state. Formally speaking, we define the *value function* of the control problem (4) as $u(x, t) = \min\limits_{m \in \mathcal{M}} J_{x,t}(m)$, where $\mathcal{M}$ denotes the set of possible control functions that we take into consideration. It can be obtained that the value function will follow a particular partial differential equation as stated below.

**Definition 3.1** ([38])**.** *The value function $u(x, t)$ is the unique solution to the following partial differential equation, which is called **Hamilton-Jacobi-Bellman Equation**:*

$$\begin{cases} \partial_t u(x, t) + \frac{1}{2} \sigma^2 \Delta u(x, t) + \min\limits_{m \in \mathcal{M}} \left[ r(x, m(t, x), t) + \nabla u \cdot m_t \right] = 0 \\ u(x, T) = g(x). \end{cases} \tag{5}$$

Hamilton-Jacobi-Bellman (HJB) equation establishes a necessary and sufficient condition for a control's optimality with regard to the cost functions. It is one of the most important high-dimensional PDEs [16] in optimal control with tremendous applications in physics [32], biology [18], and finance [26]. Many well-known equations, including Riccati equation, Linear–Quadratic–Gaussian control problem [36], Merton's portfolio problem [22] are special cases of HJB equation [37].

Conventionally, the solution to the HJB equation, i.e., the value function $u(x, t)$, can be computed using dynamic programming [2]. However, the computational complexity of dynamic programming will grow exponentially with the dimension of state function. Considering that the state function in many applications is high-dimensional, solving such HJB equations is notoriously difficult in practice using conventional solvers. As neural networks have shown impressive power in learning high-dimensional functions, it's natural to resort to neural-network-based approaches for solving high-dimensional HJB equations.

## 4 Failure Mode of PINN on High-Dimensional Stochastic Optimal Control

Note that $u(x)$ is the exact solution to the PDE (1) if and only if both loss terms $\ell_{\Omega,p}(u)$ and $\ell_{\partial\Omega,p}(u)$ are zero. However, in practice, we usually can only obtain small but non-zero loss values due to the randomness in the optimization procedure or the capacity of the neural network. In such cases, a natural question arises: whether a learned $u(x)$ with a small loss will correspond to a good approximator to the exact solution $u^*(x)$? Such a property is highly related to the concept *stability* in PDE literature, which can be defined as below in our learning scenario:

**Definition 4.1.** *Suppose $Z_1, Z_2, Z_3$ are three Banach spaces. We say a PDE defined as Eq. (1) is $(Z_1, Z_2, Z_3)$-stable, if $\|u^*(x) - u(x)\|_{Z_3} = O(\|\mathcal{L}u(x) - \varphi(x)\|_{Z_1} + \|\mathcal{B}u(x) - \psi(x)\|_{Z_2})$ as $\|\mathcal{L}u(x) - \varphi(x)\|_{Z_1}, \|\mathcal{B}u(x) - \psi(x)\|_{Z_2} \to 0$ for any function $u$.*

By definition, if a PDE is $(L^2(\Omega), L^2(\partial\Omega), Z)$-stable with a suitable Banach space $Z$, we can minimize the widely used $L^2$ Physics-Informed Losses $\|\mathcal{L}u(x) - \varphi(x)\|^2_{L^2(\Omega)}$ and $\|\mathcal{B}u(x) - \psi(x)\|^2_{L^2(\partial\Omega)}$, and the learned solution is guaranteed to be close to the exact solution when the loss terms approach zero. However, stability is not always an obvious property for PDEs. There are tremendous equations that are unstable, such as the inverse heat equation. Moreover, even if an equation is stable, it is possible that the equation is not $(L^2(\Omega), L^2(\partial\Omega), Z)$-stable, which suggests that the original $L^2$ Physics-Informed Loss might not be a good choice for solving it. We will show later that for control problems, some practical high-dimensional HJB equations are stable but not $(L^2(\Omega), L^2(\partial\Omega), Z)$-stable, and using $L^2$ Physics-Informed Loss will fail to find an approximated solution in practice.

We consider a class[3] of HJB equations in which the cost rate function is formulated as $r(x, m) = a_1|m_1|^{\alpha_1} + \cdots + a_n|m_n|^{\alpha_n} - \varphi(x, t)$. The corresponding Hamilton-Jacobi-Bellman equation can

---

[3]The form of cost function we investigate in the paper is representative in optimal control. For example, in financial markets, we often face power-law trading cost in optimal execution problems [10, 30]. The cost function in Linear–Quadratic–Gaussian control and Merton's portfolio model (constant relative risk aversion utility function in [22]) is also of this form. Therefore, we believe our theoretical analysis for this class of HJB equation is relevant for practical applications.

be reformulated as:

$$\begin{cases} \mathcal{L}_{\mathrm{HJB}}u := \partial_t u(x,t) + \frac{1}{2}\sigma^2 \Delta u(x,t) - \sum_{i=1}^{n} A_i |\partial_{x_i} u|^{c_i} = \varphi(x,t) & (x,t) \in \mathbb{R}^n \times [0,T] \\ \mathcal{B}_{\mathrm{HJB}}u := u(x,T) = g(x) & x \in \mathbb{R}^n \end{cases}, \quad (6)$$

where $A_i = (a_i \alpha_i)^{-\frac{1}{\alpha_i - 1}} - a_i(a_i \alpha_i)^{-\frac{\alpha_i}{\alpha_i - 1}} \in (0, +\infty)$ and $c_i = \frac{\alpha_i}{\alpha_i - 1} \in (1, \infty)$. See Appendix B for the detailed derivation. For a function $f : X \to \mathbb{R}$, where $X$ is a measurable space, we denote by $\mathrm{supp}f$ the support set of $f$, i.e. the closure of $\{x \in X : f(x) \neq 0\}$.

An important concept for analyzing PDEs is the Sobolev space, which is defined as follows:

**Definition 4.2.** *For $m \in \mathbb{N}$, $p \in [1, +\infty)$ and an open set $\Omega \subset \mathbb{R}^n$, the Sobolev space $W^{m,p}(\Omega)$ is defined as $\{f(x) \in L^p(\Omega) : D^\alpha f \in L^p(\Omega), \forall \alpha \in \mathbb{N}^n, |\alpha| \leq m\}$. The function space $W^{m,p}(\Omega)$ is equipped with Sobolev norm, which is defined as $\|f\|_{W^{m,p}(\Omega)} = \left( \sum_{|\alpha| \leq m} \|D^\alpha f\|_{L^p(\Omega)}^p \right)^{\frac{1}{p}}$.*

The definition above can be extended to functions defined on a spatiotemporal domain $Q \subseteq \mathbb{R}^n \times [0,T]$. With a slight abuse of notation, we define $W^{m,p}(Q) = \{f(x,t) \in L^p(Q) : D^\alpha f \in L^p(Q), \forall \alpha \in \mathbb{N}^n, |\alpha| \leq m\}$, where the differential $D^\alpha$ is only operated over spatial variable $x$. The norm $\| \cdot \|_{W^{m,p}(Q)}$ can also be defined accordingly.

**Stability of the HJB Equation** We present our main theoretical result which characterizes the stability of the HJB equation (Eq. (6)). In particular, we show that the HJB equation is $(L^p(\mathbb{R}^n \times [0,T]), L^q(\mathbb{R}^n), W^{1,r}(\mathbb{R}^n \times [0,T]))$-stable when $p$, $q$ and $r$ satisfies certain conditions. We take the Banach space $Z_3$ in Definition 4.1 as $W^{1,r}$ here because it captures the properties of both the value and the derivatives of a function, but $L^p$ spaces do not. However, as could be seen from Appendix B, for optimal control problems, it is essential to obtain an accurate approximator for both the value and the gradient of the value function $u$ (the solution of (Eq. (6))). Thus, it is appropriate to analyze the quality of the approximate solution in $W^{1,r}$ space.

**Theorem 4.3.** *For $p, q \geq 1$, let $r_0 = \frac{(n+2)q}{n+q}$. Assume the following inequalities hold for $p$, $q$ and $r_0$:*

$$p \geq \max\left\{ 2, \left(1 - \frac{1}{\bar{c}}\right)n \right\}; \quad q > \frac{(\bar{c}-1)n^2}{(2-\bar{c})n+2}; \quad \frac{1}{r_0} \geq \frac{1}{p} - \frac{1}{n}, \qquad (7)$$

*where $\bar{c} = \max_{1 \leq i \leq n} c_i$ in Eq. (6). Then for any $r \in [1, r_0)$ and any bounded open set $Q \subset \mathbb{R}^n \times [0,T]$, Eq. (6) is $(L^p(\mathbb{R}^n \times [0,T]), L^q(\mathbb{R}^n), W^{1,r}(Q))$-stable for $\bar{c} \leq 2$.*

The proof of Theorem 4.3 can be found in Appendix C and an improved theorem with relaxed dependency on $\bar{c}$ can be found in Appendix E. Intuitively, Theorem 4.3 states that $(L^p, L^q, W^{1,r})$-stability of Eq. (6) can be achieved when $p, q = \Omega(n)$. We further show that this linear dependency on $n$ cannot be relaxed in the following theorem:

**Theorem 4.4.** *There exists an instance of Eq. (6), whose exact solution is $u^*$, such that for any $\varepsilon > 0, A > 0, r \geq 1, m \in \mathbb{N}$ and $p \in \left[1, \frac{n}{4}\right]$, there exists a function $u \in C^\infty(\mathbb{R}^n \times (0,T])$ which satisfies the following conditions:*

- *$\|\mathcal{L}_{\mathrm{HJB}}u - \varphi\|_{L^p(\mathbb{R}^n \times [0,T])} < \varepsilon$, $\mathcal{B}_{\mathrm{HJB}}u = \mathcal{B}_{\mathrm{HJB}}u^*$, and $\mathrm{supp}(u - u^*)$ is compact, where $\mathcal{L}_{\mathrm{HJB}}$ and $\mathcal{B}_{\mathrm{HJB}}$ are defined in Eq. (6).*

- *$\|u - u^*\|_{W^{m,r}(\mathbb{R}^n \times [0,T])} > A$.*

The proof of Theorem 4.4 can be found in Appendix D.

**Discussion.** Theorem 4.3 and 4.4 together state that when the dimension of the state function $n$ is large, the HJB equation in Eq. (6) cannot be $(L^p, L^q, W^{1,r})$-stable if $p$ and $q$ are small. Furthermore, since $L^r = W^{0,r}$ by definition, Theorem 4.4 also implies that Eq. (6) is not even $(L^p, L^q, L^r)$-stable. Therefore, for high-dimensional HJB problems, if we use classic $L^2$ Physics-Informed Loss for training PINN, the learned solution may be arbitrarily distant from $u^*$ even if the loss is very small. Such theoretical results are verified in our empirical studies in Section 6.

More importantly, our theoretical results indicate that the design choice of the Physics-Informed Loss plays a significant role in solving PDEs using PINN. In this work, we shed light upon this problem using HJB equations. We believe the relationship between PDE's stability and the Physics-Informed Loss should be carefully investigated in the future, especially for high-dimensional non-linear PDEs whose stability are more complicated than low-dimensional and linear ones [8, 11, 19]. Given the above observations, we further propose a new algorithm for training PINN to solve HJB Equations, which will be presented in the subsequent sections.

## 5    Solving HJB Equations with Adversarial Training

The above results suggest that we should use a large value of $p$ and $q$ in the loss $\ell_{\Omega,p}(u)$ and $\ell_{\partial\Omega,q}(u)$ to guarantee a learned solution $u$ is close to $u^*$ for high-dimensional HJB problems. Note that $L^p$-norm and $L^\infty$-norm behave similarly when $p$ is large. We can substitute $L^p$-norm by $L^\infty$-norm and directly optimize $\ell_{\Omega,\infty}(u)$ and $\ell_{\partial\Omega,\infty}(u)$. Overall, the training objective can be formulated as:

$$\min_u \ \ell_\infty(u) = \sup_{x\in\Omega} |\mathcal{L}u(x) - \varphi(x)| + \lambda \sup_{x\in\partial\Omega} |\mathcal{B}u(x) - \psi(x)|, \tag{8}$$

where $\lambda > 0$ is a hyper-parameter to trade off the two objectives.

It is straightforward to obtain that setting $p$ and $q$ to infinity satisfies the conditions in Theorem 4.3, and thus the quality of the learned solution enjoys theoretical guarantee. Furthermore, Eq. (8) can be regarded as a min-max optimization problem. The inner loop is a maximization problem to find data points on $\Omega$ and $\partial\Omega$ where $u$ violates the PDE most, and the outer loop is a minimization problem to find $u$ (i.e., the neural network parameters) that minimizes the loss on those points.

In deep learning, such a min-max optimization problem has been intensively studied, and adversarial training is one of the most effective learning approaches in many applications. We leverage adversarial training, and the detailed implementation is described in Algorithm 1. In each training step, the model parameters and data points are iteratively updated. We first fix the model $u$ and randomly sample data points $x^{(1)}, \cdots, x^{(N_1)} \in \Omega$ and $\tilde{x}^{(1)}, \cdots, \tilde{x}^{(N_2)} \in \partial\Omega$, serving as a random initialization of the inner loop optimization. Then we perform gradient-based methods to obtain data points with large point-wise Physics-Informed Losses, which leads to the following inner-loop update rule:

$$x^{(k)} \leftarrow \text{Project}_\Omega \left( x^{(k)} + \eta \, \text{sign}\nabla_x \left( \mathcal{L}u_\theta(x^{(k)}) - \varphi(x^{(k)}) \right)^2 \right); \tag{9}$$

$$\tilde{x}^{(k)} \leftarrow \text{Project}_{\partial\Omega} \left( \tilde{x}^{(k)} + \eta \, \text{sign}\nabla_x \left( \mathcal{B}u_\theta(\tilde{x}^{(k)}) - \psi(\tilde{x}^{(k)}) \right)^2 \right), \tag{10}$$

---

**Algorithm 1** $L^\infty$ Training for Physics-Informed Neural Networks

---

**Input:** Target PDE (Eq. (1)); neural network $u_\theta$; initial model parameters $\theta$
**Output:** Learned PDE solution $u_\theta$
**Hyper-parameters:** Number of total training iterations $M$; number of iterations and step size of inner loop $K, \eta$; weight for combining the two loss term $\lambda$

1: **for** $i = 1, \cdots, M$ **do**
2:     Sample $x^{(1)}, \cdots, x^{(N_1)} \in \Omega$ and $\tilde{x}^{(1)}, \cdots, \tilde{x}^{(N_2)} \in \partial\Omega$
3:     **for** $j = 1, \cdots, K$ **do**
4:         **for** $k = 1, \cdots, N_1$ **do**
5:             $x^{(k)} \leftarrow \text{Project}_\Omega \left( x^{(k)} + \eta \, \text{sign}\nabla_x \left( \mathcal{L}u_\theta(x^{(k)}) - \varphi(x^{(k)}) \right)^2 \right)$
6:         **for** $k = 1, \cdots, N_2$ **do**
7:             $\tilde{x}^{(k)} \leftarrow \text{Project}_{\partial\Omega} \left( \tilde{x}^{(k)} + \eta \, \text{sign}\nabla_x \left( \mathcal{B}u_\theta(\tilde{x}^{(k)}) - \psi(\tilde{x}^{(k)}) \right)^2 \right)$
8:     $g \leftarrow \nabla_\theta \left( \dfrac{1}{N_1} \sum_{i=1}^{N_1} \left( \mathcal{L}u_\theta(x^{(i)}) - \varphi(x^{(i)}) \right)^2 + \lambda \cdot \dfrac{1}{N_2} \sum_{i=1}^{N_2} \left( \mathcal{B}u_\theta(\tilde{x}^{(i)}) - \psi(\tilde{x}^{(i)}) \right)^2 \right)$
9:     $\theta \leftarrow \text{Optimizer}(\theta, g)$
10: **return** $u_\theta$

---

Table 1: **Experimental results of solving the 100/250-dimensional LQG control problems.** $n$ denotes the dimensionality of the problem. Performances are measured by $L^1$, $L^2$, and $W^{1,1}$ relative error in $[0,1]^n \times [0,T]$. The best performances are indicated in **bold**.

| Method | Relative error for $n = 100$ | | | Relative error for $n = 250$ | | |
|---|---|---|---|---|---|---|
| | $L^1$ | $L^2$ | $W^{1,1}$ | $L^1$ | $L^2$ | $W^{1,1}$ |
| Original PINN [28] | 3.47% | 4.25% | 11.31% | 6.74% | 7.67% | 17.51% |
| Adaptive time sampling [35] | 3.05% | 3.67% | 13.63% | 7.18% | 7.91% | 18.38% |
| Learning rate annealing [34] | 11.09% | 11.82% | 33.61% | 6.94% | 8.04% | 18.47% |
| Curriculum regularization [17] | 3.40% | 3.91% | 9.53% | 6.72% | 7.51% | 17.52% |
| Adversarial training (ours) | **0.27%** | **0.33%** | **2.22%** | **0.95%** | **1.18%** | **4.38%** |

where $\text{Project}_\Omega(\cdot)$ and $\text{Project}_{\partial\Omega}(\cdot)$ project the updated data points to the domain. When the inner-loop optimization finishes, we fix the generated data points and calculate the gradient $g$ to the model parameter:

$$g \leftarrow \nabla_\theta \left( \frac{1}{N_1} \sum_{i=1}^{N_1} \left( \mathcal{L}u_\theta(x^{(i)}) - \varphi(x^{(i)}) \right)^2 + \lambda \cdot \frac{1}{N_2} \sum_{i=1}^{N_2} \left( \mathcal{B}u_\theta(\tilde{x}^{(i)}) - \psi(\tilde{x}^{(i)}) \right)^2 \right), \quad (11)$$

then the model parameter can be updated using any first-order optimization methods. When the training finishes, the learned neural network will converge to a solution with small $L^\infty$ losses and is guaranteed to be close to the exact solution.

## 6  Experiments

In this section, we conduct experiments to verify the effectiveness of our approach. Ablation studies on the design choices and hyper-parameters are then provided. Our codes are implemented based on `PyTorch` [23]. All the models are trained on one NVIDIA Tesla V100 GPU with 16GB memory. Due to space limitation, we only showcase our methods on the Linear Quadratic Gaussian control problem in the main body of the paper. More experimental results on other PDE problems can be found in Appendix G.

### 6.1  High Dimensional Linear Quadratic Gaussian Control Problem

We follow [13] to study the classical linear-quadratic Gaussian (LQG) control problem in $n$ dimensions, a special case of the HJB equation:

$$\begin{cases} \partial_t u(x,t) + \Delta u(x,t) - \mu\|\nabla_x u(x,t)\|^2 = 0 & x \in \mathbb{R}^n, t \in [0,T] \\ u(x,T) = g(x) & x \in \mathbb{R}^n, \end{cases} \quad (12)$$

As is shown in [13], there is a unique solution to Eq. (12):

$$u(x,t) = -\frac{1}{\mu} \ln \left( \int_{\mathbb{R}^n} (2\pi)^{-n/2} e^{-\|y\|^2/2} \cdot e^{-\mu g(x - \sqrt{2(T-t)}y)} \mathrm{d}y \right), \quad (13)$$

We set $\mu = 1$, $T = 1$, and the terminal cost function $g(x) = \ln \left( \frac{1 + \|x\|^2}{2} \right)$.

**Experimental Design**  The neural network used for training is a 4-layer MLP with 4096 neurons and $\tanh$ activation in each hidden layer. To train the models, we use Adam as the optimizer [15]. The learning rate is set to $7e-4$ in the beginning and then decays linearly to zero during training. The total number of training iterations is set to 5000/10000 for the 100/250-dimensional problem. In each training iteration, we sample $N_1 = 100/50$ points from the domain $\mathbb{R}^n \times [0,T]$ and $N_2 = 100/50$ points from the boundary $\mathbb{R}^n \times \{T\}$ to obtain a mini-batch for the 100/250-dimensional problem. The number of inner-loop iterations $K$ is set to 20, and the inner-loop step size $\eta$ is set to 0.05 unless otherwise specified. Evaluations are performed on a hold-out validation set which is unseen during

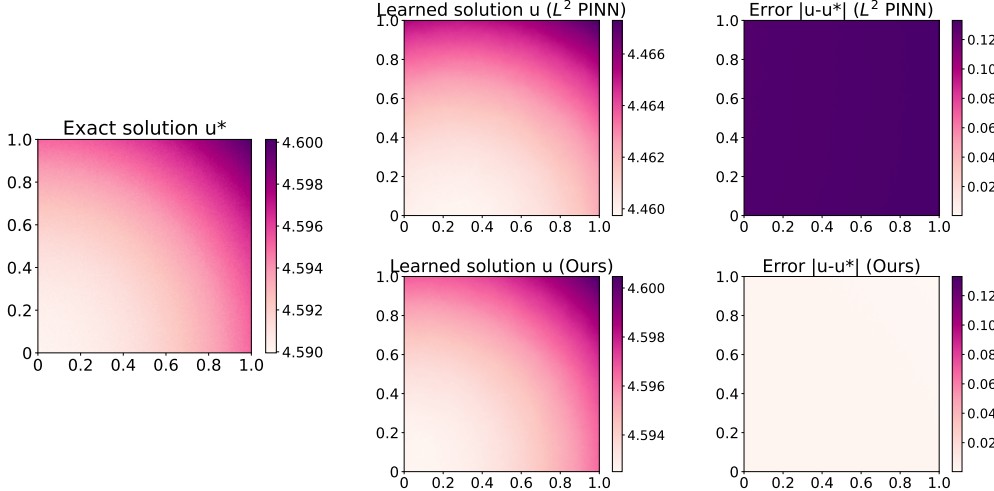

Figure 1: **Visualization for the solutions of Eq. (12)**. The left panel shows the exact solution $u^*$; the middle panel shows the learned solutions $u$ of the original PINN method with $L^2$ loss and our method with adversarial training; the right panel show the point-wise absolute error $|u - u^*|$. Note that the solution is a high dimensional function, and we visualize its snapshot on a two-dimensional domain. Specifically, we visualize a bivariate function $u(x_1, x_2, 0, \cdots, 0; 0)$ for $x_1, x_2 \in [0, 1]$ with the horizontal axis and vertical axis corresponding to $x_1$ and $x_2$ respectively.

training. We use the $L^1$, $L^2$, and $W^{1,1}$ relative error in $[0, 1]^n \times [0, T]$ as evaluation metrics: $L^1$ and $L^2$ relative errors are popular evaluation metrics in literature. We additionally consider $W^{1,1}$ relative error since the gradient of the solution to HJB equations plays an important role in applications, and our theory indicates that Eq. (12) is $(L^\infty, L^\infty, W^{1,r})$ stable. More detailed descriptions of the experimental setting and evaluation metrics can be found in Appendix F.

We compare our method with a few strong baselines: 1) original PINN trained with $L^2$ Physics-Informed Loss [28]; 2) adaptive time sampling for PINN training proposed in [35]; 3) PINN with the learning rate annealing algorithm proposed in [34]; 4) curriculum PINN regularization proposed in [17]. The training recipes for the baseline methods, including the neural network architecture, the training iterations, the optimizer, and the learning rate, are the same as those of our method described above. It should be noted that although these approaches modifies the data sampler, training algorithms or the loss function, they all keep the $L^2$ norm of the PDE residual and boundary residual unchanged in the training objective.

**Experimental Results**  The experimental results are summarized in Table 1. It's clear that the relative error of the model trained using the original PINN does not fit the solution well, e.g., the $L^1$ relative error is larger than $6\%$ when $n = 250$. This empirical observation aligns well with our theoretical analysis, i.e., minimizing $L^2$ loss cannot guarantee the learned solution to be accurate. Advanced methods, e.g., curriculum PINN regularization [17], can improve the accuracy of the learned solution but with marginal improvement, which suggests that these methods do not address the key limitation of PINN in solving high-dimensional HJB Equations. By contrast, our proposed method significantly outperforms all the baseline methods in terms of both $L^p$ relative error and Sobolev relative error, which indicates that both the values and the gradients of our learned solutions are more accurate than the baselines.

We also examine the quality of the learned solution $u(x, t)$ by visualization. As the solution is a high-dimensional function, we visualize its snapshot on a two-dimensional space. Specifically, we consider a bivariate function $u(x_1, x_2, 0, \cdots, 0; 0)$ and use a heatmap to show its function value given different $x_1$ and $x_2$. Figure 1 shows the ground truth $u^*$, the learned solutions $u$ of original PINN and our method, and the point-wise absolute error $|u - u^*|$ for each methods. The two axises correspond $x_1$ and $x_2$, respectively. We can see that the point-wise error of the learned solution using our algorithm is less than $2\mathrm{e} - 2$ on average. In contrast, the point-wise error of the learned solution using original PINN method with $L^2$ loss is larger than $1.3\mathrm{e} - 1$ for most areas. Therefore,

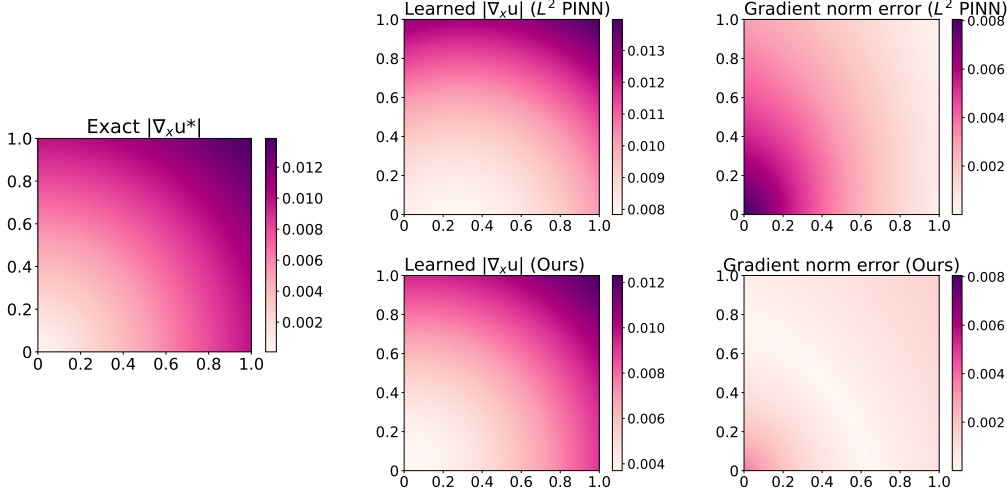

Figure 2: **Visualization for the gradient norm of the solutions of Eq. (12)**. The left panel shows the ground truth $|\nabla_x u^*|$; the middle panel shows the learned $|\nabla_x u|$ of the original PINN method with $L^2$ loss and our method with adversarial training; the right panel shows the point-wise absolute error $|\nabla_x(u - u^*)|$. We only visualize a snapshot on a two-dimensional domain. Specifically, we visualize a bivariate function $u(x_1, x_2, 0, \cdots, 0; 0)$ for $x_1, x_2 \in [0, 1]$ with the horizontal axis and vertical axis corresponding to $x_1$ and $x_2$ respectively.

the visualization of the solutions clearly illustrate that PINN learned more accurate solution using our proposed algorithm.

Furthermore, we visualize the gradient norm $|\nabla_x u|$ of the learned solution of both our method and the original PINN in Figure 2, to illustrate that not only can the learned solution of our method accurately approximate the exact solution, but also the *gradient* of the learned solution can accurately approximate the *gradient* of the exact solution. Again, since $|\nabla_x u|$ is a high dimensional function, we use a heatmap to show its function value given different $x_1$ and $x_2$, and set the other variables to 0. From the right panel of Figure 2, we can clearly see that the gradient of the learned solution of our method is much more accurate compared with that of the gradient of the learned solution using original PINN. The gradient norm error of the vanilla PINN approach is nearly $1e-2$ in some areas shown in the visualization, while the error of our method is less than $1e-3$ for most data points. This empirical observation aligns well with our theory, which states that Eq. (12) is $(L^\infty, L^\infty, W^{1,r})$ stable.

## 6.2 Ablation studies

We conduct ablation studies on the 100-dimensional LQG control problem (Eq. (12)) to ablate the main designs in our algorithm.

**Adversarial training v.s. Directly optimizing $L^p$ physic-informed loss.** Our proposed algorithm introduces a min-max optimization procedure. One may have concerns that such an approach may be unnecessarily complicated, and directly minimizing $L^p$ physic-informed loss with a large $p$ would have the same effect. We use these two methods to solve Eq. (12), and compare their performance in the left panel of Table 2. It can be seen that directly minimizing $L^p$ physic-informed loss does not lead to satisfactory results.

We point out that this observation does not contradict our theoretical analysis (Theorem 4.3). Theorem 4.3 focuses on the *approximation* ability, which indicates that a model with a small $L^p$ loss can approximate the exact solution well. The empirical results in Table 2 demonstrate the *optimization* difficulty of learning such a model with $L^p$ loss. By comparison, our proposed adversarial training method is more stable and leads to better performance. More detailed discussions are provided in Appendix H.

Table 2: **Experimental results for ablation studies.** The left panel compares PINN trained with $L^p$ Physic Informed Loss and our method; the right panel compares PINN trained with partial or no adversarial training and our method. In the first two columns of the right panel, ✗ indicates using the original $L^2$ Physics-Informed Loss for the PDE/boundary residual loss term, while ✓ indicates using the proposed adversarial training method for the corresponding loss term. Performances are measured by $L^1$ relative error. Best performances are indicated in **bold**.

| Method | Relative error |
|---|---|
| $L^4$ Loss | 2.42% |
| $L^8$ Loss | 53.55% |
| $L^{16}$ Loss | 113.24% |
| Ours | **0.27%** |

| Adversarial training | | Relative error |
|---|---|---|
| Domain | Boundary | |
| ✗ | ✗ | 3.47% |
| ✗ | ✓ | 2.79% |
| ✓ | ✓ | **0.27%** |

$L^1$ relative error (%)

| | $\eta = 0.05$ | $\eta = 0.1$ | $\eta = 0.2$ |
|---|---|---|---|
| $K = 5$ | 1.7 | 0.97 | 0.49 |
| $K = 10$ | 1.6 | 0.46 | 2.5 |
| $K = 20$ | 0.27 | 1.2 | 8.8 |

**Adversarial training should be applied to both the PDE residual and the boundary residual.**
Our theoretical analysis suggests that we should use a large value of $p$ and $q$ in the loss $\ell_{\Omega,p}(u)$ and $\ell_{\partial\Omega,q}(u)$ to guarantee the quality of the learned solution $u$. Thus, in the proposed Algorithm 1, both the data points inside the domain and the data points on the boundary are learned in the inner-loop maximization. From the right panel of Table 2, we can see that when adversarial training is applied to one loss term, the performance is slightly improved, but its accuracy is still not satisfactory. When both loss terms use adversarial training, the solution is one order of magnitude more accurate, indicating that applying adversarial training to the whole loss function is essential.

**Hyper-parameters $K$ and $\eta$ for the inner loop maximization.** Our approach introduces additional hyper-parameters $K$ (the number of inner-loop iterations) and $\eta$ (the inner-loop step size). These two parameters control the accuracy of inner-loop maximization. We conduct ablation studies to examine the effects of different design choices. Specifically, we experiment with $K = 5, 10, 20$ and $\eta = 0.05, 0.1, 0.2$, and show the $L^1$ relative error in the right panel of Table 2. Typically we find that setting the product $K\eta = 1$ achieves the best performance. When $K\eta$ is fixed, our results suggest that using a larger $K$ and a smaller $\eta$, i.e., more inner-loop iterations and smaller step sizes, will lead to better performance while being more time-consuming.

## 7 Conclusions

In this paper, we theoretically investigate the relationship between the loss function and the approximation quality of the learned solution using the concept of stability in the partial differential equation. We study an important class of high-dimensional non-linear PDEs in optimal control, the Hamilton-Jacobi-Bellman (HJB) equation, and prove that for general $L^p$ Physics-Informed Loss, the HJB equation is stable only if $p$ is sufficiently large. Such a theoretical finding reveals that the widely used $L^2$ loss is not suitable for training PINN on high-dimensional HJB equations, while $L^\infty$ loss is a better choice. The theory also inspires us to develop a novel PINN training algorithm to minimize the $L^\infty$ loss for HJB equations in a similar spirit to adversarial training. One limitation of this work is that we only work on the HJB Equation. Theoretical investigation of other important equations can be an exciting direction for future works. We believe this work provides important insights into the loss design in Physics-Informed deep learning.

## Acknowledgements

We thank Weinan E, Bin Dong, Yiping Lu, Zhifei Zhang, and Yufan Chen for the helpful discussions.

This work is supported by National Science Foundation of China (NSFC62276005), The Major Key Project of PCL (PCL2021A12), Exploratory Research Project of Zhejiang Lab (No. 2022RC0AN02), and Project 2020BD006 supported by PKUBaidu Fund.

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
