# OpenReview forum: "Is $L^2$ Physics Informed Loss Always Suitable for Training Physics Informed Neural Network?"
_NeurIPS.cc/2022/Conference — NeurIPS 2022 Accept_

### Official Review · Reviewer_bGoA · 2022-07-11

**Rating:** 6
**Confidence:** 3
**Soundness:** 3 good
**Presentation:** 3 good
**Contribution:** 3 good

**Summary:**

Te present work is concerned with the application of physics-informed neural networks (PINNS) to the solution of high-dimensional Hamilton-Jacobi-Bellman equations (HJB). They show that in a wide range of cases directly applying the standard squared penalty to HJB and boundary conditions does not result in a stable problem, meaning that the PINN loss can go to zero without the PINN solution approximating the true solution. In order to overcome this problem, the authors propose to use the higher-order $L^p$ penalties. They show that higher-order $L^p$ penalties achieve stability under a wider range of conditions, with $L^\infty$ being the extreme case. They then design an adversarial training procedure to train with the $L^\infty$ loss in practice and show that it improves the relative accuracy of the solution compared to existing PINN variants.

### After rebuttal:
The authors have addressed my concerns appropriately. I now recommend acceptance.

**Questions:**

1. In the theorems 4.3 and 4.4, the target accuracy is measured in the $W^{1,r}$ norm, meaning in terms of the $L^r$ norm of both the solution and its gradient. However, in the experiments, the error is measured in terms of just the $L^p$ norms of the solution. Thus, experiments do not actually seem to illustrate the theoretical results as advertised.

2. Following up on 1, are similar results true when measuring solution accuracy in $L^p$ spaces (instead of sobolev spaces)? If not, the authors should provide more motivation accuracy in sobolev spaces is important.

3. Did you find that different learning rate schedules had an effect on the accuracy of Adam?

4. The error plots of Figure 4 are not very meaningful instead, or complementing them I think it would be important to show learning to show time series of the training and test error. This would allow to distinguish difficulties of the finite batch setting to accurately approximate the infinite-batch-training loss from difficulties in optimizing the finite batch loss, from the problem of stability that is motivating the author's technique.


**Limitations:**

As discussed above under "questions", there presently seem to be gaps between theory and experiments that are not commented on in the text.

**Strengths And Weaknesses:**

Strengths: The method is well-motivated and seems to perform well in practice on the problem of interest.

Weakness: As detailed under "questions", the present version of the paper leaves doubts regarding some of the claims. The experimental validation seems to be in a setting not covered by the theory and the most common diagnostics, such as error vs time, are missing.

I want to emphasize that I think that this has the potential to be a solid paper suitable for acceptance. I am giving a somewhat lower score right now but hope to be able to raise it if the authors can clarify my concerns.

---

> ### Author Response · Authors · 2022-08-01
> **Response to Reviewer bGoA**
>
>
> Thanks for your careful review! We respond to your questions as below.
>
> **Response to Question 1&2.**
> It is a good suggestion. We have examined the quality of the gradient of the learned solution in Figure 3, Appendix G. Following your advice, we further conduct experiments to compute $L^2$ and $W^{1,1}$ relative errors of our model and the baseline methods on the 100-dimensional HJB Equation (Eq.(12)). The results are shown below:
>
> |                           | $L^1$         | $L^2$         | $W^{1,1}$     |
> | ----                      | ----          | ----          | ----          |
> | Original PINN             | 3.47%      | 4.25%      |  11.31%    |
> | Adaptive time sampling    | 3.05%      | 3.67%      |  13.63%    |
> | Learning rate annealing   | 11.09%     | 11.82%     |  33.61%    |
> | Curriculum regularization | 3.40%      | 3.91%      |  9.53%     |
> | Ours                      | **0.27**%  | **0.33**%  |  **2.22**% |
>
> Clearly, our approach significantly outperforms the baselines in terms of both $L^p$ norms and Sobolev norms. Indeed, the Sobolev norm is stronger in the sense that if a PDE is $(L^p, L^q, W^{1,r})$-stable, then it must be $(L^p, L^q, L^{r})$-stable by definition. We will include these results in the paper revision to make our claims more convincing.
>
> **Response to Question 3.**
> We follow the standard practice in other fields such as NLP and CV [BERT, RoBERTa and Vision Transformers] to use linear learning rate decay (i.e., decrease the learning rate linearly to 0 during training) for all experiments, including baselines and our models (see Appendix F). This strategy has been shown to lead to more effective optimization than using a constant learning rate.
>
> **Response to Question 4.**
> Thanks for the suggestion! We will include error/loss-vs-time plots and add some corresponding discussions in the paper revision. Since it's hard to include figures in the response, we present our error/loss-vs-time result in the following tables:
>
> Error/loss-vs-time result of original PINN
> | iteration                | 1000   | 2000   | 3000   | 4000   | 5000   |
> |--------------------------|--------|--------|--------|--------|--------|
> | $L^2$ loss               | 0.098  | 0.088  | 0.070  | 0.584  | 0.041  |
> | $L^1$ relative error     | 6.18%  | 5.36%  | 3.86%  | 3.94%  | 3.47%  |
> | $W^{1,1}$ relative error | 17.53% | 17.67% | 14.83% | 14.40% | 11.31% |
>
>
> Error/loss-vs-time result of our method
> | iteration                | 1000   | 2000   | 3000   | 4000   | 5000   |
> |--------------------------|--------|--------|--------|--------|--------|
> | $L^{\infty}$ loss        | 11.841 | 9.352  | 2.404  | 1.605  | 0.711  |
> | $L^1$ relative error     | 15.22% | 4.26%  | 0.97%  | 1.10%  | 0.27%  |
> | $W^{1,1}$ relative error | 21.91% | 18.62% | 5.14%  | 4.96%  | 2.22%  |
>
> It's clear that for the original PINN approach, the $L^2$ loss drops very quickly during training, while its $W^{1,1}$ relative error remains high. This result indicates the optimization is successful in this experiment, and that the stability property of the PDE leads to the high test error. By contrast, our proposed training approach enables the test error goes down steadily during training, which aligns with the theoretical claims.
>
> We sincerely hope that our responses address your concerns and you can reevaluate the quality of our submission. We are also willing to discuss with you if you have any further questions.

---

> > ### Comment · Reviewer_bGoA · 2022-08-07
> > **One last question**
> >
> > Thanks, for addressing my concerns! One follow-up on question 2: My question was more pointed toward whether the negative results on regular PINNs also hold when targeting accuracy in $L^p$ spaces instead of Sobolev spaces or why one should care about the accuracy in Sobolev norms instead of p-norms. Put another way: Why should one care about your theory on accuracy in Sobolev norms, given that most often, accuracy in p-norms is reported?

---

> > > ### Author Response · Authors · 2022-08-08
> > > **Response to the follow-up question**
> > >
> > > Thanks for your feedback! We notice that your concern are two-fold: whether the negative results of PINN holds when the error is measured with the $L^p$ norm and why we investigate the setting under the Sobolev norm.
> > >
> > > **Regarding the first concern.**
> > > Theorem 4.4 has already indicated that vanilla PINN will fail when the error is measured in $L^p$ norm. To be specific, since $L^p$ norm is a special case of Sobolev norm ($L^p=W^{0,p}$ by definition), we can set $m=0$ in Theorem 4.4 and obtain that for any $p\geq 1$, $\\|u^*-u_\theta\\|_p$ could be arbitrarily large if the approximator $u_\theta$ is learned with $L^2$ loss.
> > >
> > > **Regarding the second concern.**
> > > In many practical PDE problems, people not only care about the approximation error of $u$ but also $\nabla u$. Therefore, the Sobolev norm is a more proper metric than the $L^p$ norm in PDE literature [1-3] since it captures the properties of both the value and the derivatives of a function.
> > >
> > > To be concrete, for our study on HJB equations, as is shown in Remark B.2 in Appendix B, $\nabla u^*$ ($u^*$ denotes the exact solution) is of great significance in application since it's closely related to the optimal control function. Therefore, it is essential to obtain an accurate approximator for both $u^*$ and $\nabla u^*$, which the Sobolev norm can precisely characterize.
> > >
> > > We hope our explanations can address your concern and you can re-evaluate our work based on that.
> > >
> > > [1] Evans L C. Partial differential equations[M]. American Mathematical Soc., 2010.
> > >
> > > [2] Gilbarg D, Trudinger N S, Gilbarg D, et al. Elliptic partial differential equations of second order[M]. Berlin: springer, 1977.
> > >
> > > [3] Lieberman G M. Second order parabolic differential equations[M]. World scientific, 1996.

---

> > > > ### Comment · Reviewer_bGoA · 2022-08-08
> > > > **Thanks for the explanation, I will update my score**
> > > >
> > > > Thanks for the explanation, I suggest making some of these more concrete in the paper. Meanwhile, I have updated my score.

---

> > > > > ### Author Response · Authors · 2022-08-08
> > > > > **Thank you for the rating update!**
> > > > >
> > > > > Thank you very much for your re-evaluation and rating update. We will include the relevant discussions on Sobolev norms and $L^p$ norms in the next version of our paper.

---

### Official Review · Reviewer_a7nv · 2022-07-12

**Rating:** 7
**Confidence:** 3
**Soundness:** 3 good
**Presentation:** 4 excellent
**Contribution:** 3 good

**Summary:**

This paper studies the choice of loss functions when using neural networks to solve partial differential equations. The authors establish positive and negative results showing the conditions under which the problem of solving a Hamilton-Jacobi-Bellman equation is stable/unstable. Based on this, an $L^\infty$ training approach is proposed. Experiments demonstrate that the proposed method has better performance than existing methods.


**Questions:**

I suggest that the authors could address the weaknesses mentioned in the strengths and weaknesses section.

**Limitations:**

This paper does not have any potential negative societal impact.

**Strengths And Weaknesses:**

Strengths:

The results of this paper are solid and comprehensive.

The paper is well-organized and easy to follow.

The experiments clearly demonstrate the advantage of the proposed $L^\infty$ loss over the original PINN methods.



Weaknesses:

This work focuses on a quite specific class of HJB equations and requires $\bar{c} \leq 2$ for the stability result to hold, which seems to require $\alpha_i \geq 2$ in the cost rate function.

Table 2 in the ablation studies seems to contradict the theorem results. Some more discussions may be quite helpful here to explain why finite but larger values of p lead to worse performances.

---

> ### Author Response · Authors · 2022-08-01
> **Response to Reviewer a7nv**
>
> Thank you very much for supporting our work. We respond to each of your concerns as below.
>
> **Regarding the focus of our work.**
> This work is the first to rigorously demonstrate that choosing a proper loss function is critical for some practical (and important) PDEs. We believe the loss function design in PINN is under-explored and agree with the reviewer that the current theoretical analysis can be extended from many aspects, such as relaxing the assumptions and studying other PDEs. We will leave them as future work and provide a deeper understanding of PINN loss and differential equations.
>
> **Regarding contradiction between theorem and large $p$ training.**
> There is no contradiction between our theorems and empirical results. Theorem 4.3 focuses on the approximation ability, which indicates that if we have a model whose $L^p$ loss is small, it will approximate the true solution well. The empirical results in Table 2 demonstrate the optimization difficulty of learning such a model.
> Intuitively, we randomly sample points in each training iteration in the domain/boundary to calculate the loss. When $p$ is large, most sampled points will hardly contribute to the loss, which leads to inefficiency and makes the training hard to converge. In Algorithm 1, we adversarially learn the points with large loss values, making all of them contribute to the model update (Step 8), significantly improving the model training.
>
> Technically, directly applying Monte Carlo to compute $L^p$ loss in experiments will lead to large variance estimations. For a function $f$,
> $$
>     \int |f|^p \mathrm{d}x=\frac 1 N \sum_{i=1}^N |f(X_i)|^p+O\left(\sqrt{\frac{\mathrm{Var} |f(X)|^p}{N}}\right).
> $$
>
> Thus, $||f||_p$ suffers from an $O((\mathrm{Var} |f(X)|^p/N)^{1/2p})$ error.
>
> As $p\to\infty, \mathrm{Var} |f(X)|^p\sim ||f||_{\infty}^{2p}$. Therefore, the errors for estimating both Eq.(2,3) and the $L^p$ norm of the residual are very large when $p$ is large.
>
> We appreciate your question which helps us realize the problem in the presentation. We will revise the paper accordingly.
>
> May you have any further questions, please tell us and we are willing to address your concerns.

---

### Official Review · Reviewer_RgXD · 2022-07-12

**Rating:** 6
**Confidence:** 4
**Soundness:** 4 excellent
**Presentation:** 4 excellent
**Contribution:** 3 good

**Summary:**

This paper studies the objective function in physics-informed learning for HJB equations. Authors propose the concept of stability to analyze what is a good choice of loss function. The theoretical result challenges the common practice of $L^2$ and suggests that $L^p$ with large $p$ gives stability guarantee. The authors propose an adversarial training algorithm to minimize $L^{\infty}$ loss based on their theoretical finding.  The proposed method demonstrates superior empirical performance in the simulated high-dimensional problem.

**Questions:**

1. In the stochastic differential equation given by Eq (4), the output of control function has the same dimension as the state. Also, the control function only depends on time. But I think the control variable usually has different dimension from the state variable and the control function should be a function of both time and state. The setting described in Eq (4) seems too simple compared to what I found in the literature equation 2.10 in section 2.4 [1].  Is this a standard setting in optimal control?
2. Given the results in Table 2 and what authors said in page 5 -"$L^p$ and $L^{\infty}$ -norm behave similarly when $p$ is large", it seems that minimizing the exact $L^{\infty}$ loss may have poor performance. The reason why the adversarial training works better than $L^2$ seems to be that adversarial training fails to approximate $L^{\infty}$ -norm well. Is it possible to check
3. Given the training objective in Eq (8), shouldn't the objective functions in algorithm line 5,7,8 use absolute value instead of $L^2$?

[1] Lu, Q., and Xu Zhang. "A mini-course on stochastic control." _Control and inverse problems for partial differential equations_ 22 (2016): 171-254.

**Limitations:**

Listed in the weakness and question sections.

**Strengths And Weaknesses:**

Strengths:
1. This paper is well-motivated and well-written. Stability analysis is important for using physics-informed learning to solve PDEs.
2. The theoretical results advance the understanding of loss design in physics-informed learning.
3. The proposed method demonstrates superior empirical performance compared to other baselines in the simulated high-dimensional problem.

Weakness:
1. Adversarial training is time consuming and it introduces more hyperparameters to tune.
2. The experiment only considers two special cases (including the one in the appendix) of HJB equations which have close form solutions. The one in the appendix doesn't compare with other baselines. The empirical evidence is not strong enough to me.
3. The theoretical results suggest Lp loss with a large p guarantees better stability than L2. However, the results in Table 2 turn out to be the opposite. Explanation in the Section 6.2 seems vague.
4. The setting seems restricted. More details are provided in the questions.

---

> ### Author Response · Authors · 2022-08-01
> **Response to Reviewer RgXD**
>
> Thanks for your careful review! We respond to your concerns as below.
>
> **Response to Weakness 1.**
> Thanks for the question. In this work, we focus on learning an accurate solution. We prove that for some PDEs, accurate solutions cannot be learned by minimizing $L^2$ loss although the training is faster. We agree that adversarial training needs more computations and introduces two additional hyper-parameters. This problem has already been tackled by several recent works in the field of adversarial robustness [1-2]. We will try those efficient and robust adversarial training in our problem and explore more in this direction.
>
> [1] Wong, Eric, Leslie Rice, and J. Zico Kolter. "Fast is better than free: Revisiting adversarial training." International Conference on Learning Representations. 2019.
>
> [2] Zhang, Dinghuai, et al. "You only propagate once: Accelerating adversarial training via maximal principle." Advances in Neural Information Processing Systems 32 (2019).
>
> **Response to Weakness 2.**
> Thanks for pointing out the problem and suggestion. Note that not all HJB equations have analytical solutions, with which we can compare the approximation quality of different algorithms. Therefore, we follow previous works and select several equations with closed-form solutions for evaluation. As for the lack of comparison with other baselines, we follow your advice to further conduct experiments on the equations in the appendix using other baseline methods. The relative errors are shown below:
>
> |                           | $c=1.25$      | $c=1.5$       | $c=1.75$      |
> | ----                      | ----          | ----          | ----          |
> | Original PINN             | 1.11%      | 3.82%      |  2.73%     |
> | Adaptive time sampling    | 1.18%      | 2.34%      |  7.94%     |
> | Learning rate annealing   | 0.98%      | 1.13%      |  1.06%     |
> | Curriculum regularization | 6.27%      | 0.37%      |  3.51%     |
> | Ours                      | **0.61**%  | **0.15**%  |  **0.29**% |
>
> It's clear that our approach outperforms other baselines in all these equations. We will include these results in the paper revision.
>
> **Response to Weakness 3 and Question 2.**
> There is no contradiction between our theorems and empirical results. Theorem 4.3 focuses on the approximation ability, which indicates that if we have a model whose $L^p$ loss is small, it will approximate the true solution well. The empirical results in Table 2 demonstrate the optimization difficulty of learning such a model.
> Intuitively, we randomly sample points in each training iteration in the domain/boundary to calculate the loss. When $p$ is large, most sampled points will hardly contribute to the loss, which leads to inefficiency and makes the training hard to converge. In Algorithm 1, we adversarially learn the points with large loss values, making all of them contribute to the model update (Step 8), significantly improving the model training.
>
> Technically, directly applying Monte Carlo to compute $L^p$ loss in experiments will lead to large variance estimations. For a function $f$,
> $$
>     \int |f|^p \mathrm{d}x=\frac 1 N \sum_{i=1}^N |f(X_i)|^p+O\left(\sqrt{\frac{\mathrm{Var} |f(X)|^p}{N}}\right).
> $$
>
> Thus, $||f||_p$ suffers from an $O((\mathrm{Var} |f(X)|^p/N)^{1/2p})$ error.
>
> As $p\to\infty, \mathrm{Var} |f(X)|^p\sim ||f||_{\infty}^{2p}$. Therefore, the errors for estimating both Eq.(2,3) and the $L^p$ norm of the residual are very large when $p$ is large.
>
> We appreciate your question which helps us realize the problem in the presentation. We will revise the paper accordingly.
>
> **Response to Weakness 4 and Question 1.**
> Thanks for the question. We kindly point out that the equation we target in the paper is general and can cover settings regarding state/control dimension mismatch and state-dependent control function by some simple reformulation. For example, the dimension mismatch issue can be solved by simply adding 0 to the vector with smaller dimensionality, e.g., changing 3-dimensional control $(0.1,0.2,-0.1)$ to $(0.1,0.2,-0.1,0,0)$ which is a 5-dimentional vector. Following the derivation in Chapter 3 of reference [33], we could also reformulate the HJB equation of an optimal control problem involving state-dependent control functions into a (same) PDE only involving state-independent control functions.
>
> **Response to Question 3.**
> We thank the reviewer for pointing out this. The absolute value should be a better surrogate for the loss in Algorithm 1. But practically $L^2$ loss leads to a smoother gradient, and further experiments verify that this choice has little impact on the performance, with $L^2$ loss being slightly better.
>
> May you have any further questions, please tell us and we are willing to address your concerns.

---

> > ### Comment · Reviewer_RgXD · 2022-08-05
> > **Thanks for the response**
> >
> > Thank authors for the detailed response. Most of my concerns have been addressed. I'll update my rating. As for reformulating HJB equation with state-dependent control function into PDE with state-independent control function, I would suggest authors include the derivation and discussion in the appendix because it does not seem obvious.

---

> > > ### Author Response · Authors · 2022-08-06
> > > **Thanks you for the quick response and rating update!**
> > >
> > > We are delighted to see that our response has addressed your concerns. We will follow your suggestions to add more derivations and discussions in the appendix to illustrate the generality of the equation we consider.

---

### Official Review · Reviewer_hCTD · 2022-07-13

**Rating:** 6
**Confidence:** 4
**Soundness:** 3 good
**Presentation:** 2 fair
**Contribution:** 3 good

**Summary:**

The paper questions the status quo of using L2-loss for optimizing PINNs. Theoretical results show that a learned solution to HJB is (Z1, Z2, Z3)-stable iff the physics-informed loss uses Lp loss with sufficiently large p. Empirical results partially support the theory and show that optimizing L-inf loss will result in lower L1 error in the case of a linear LQG problem. As L-inf loss is challenging to optimize with Adam, the paper proposes an adversarial training algorithm that indeed outperforms optimization of L-inf with Adam.


**Questions:**

Results and Impact:
1) The authors evaluate the approximation quality of their method on L1 error (Table 1). What is the justification for only using L1 to empirically define a "good approximator of the exact solution" (L34)? Indeed, the proposed L-inf loss results in a lower L1 error than L2 loss on a high-dim. linear LQG problem. But, it is unclear to me if L-inf loss will also result in a lower L2 error. I would be willing to raise my score with a comparison of L2 error and a discussion of it.
2) Can a general statement be made that L-inf loss will be a more appropriate loss choice for high-dimensional problems?

Related Works:

3) What is the broader novelty wrt. prior works that use deep learning for HJB? The related works state that there exists "several works" that solve HJB with deep learning, yet the authors only mention two works [L82-90]. A quick Google shows that there is other works by, e.g., Nüsken and Richter, 2021;
4) There exist works that use GANs instead of L2 loss. What is the advantage of using L-inf over the GAN loss?
5) Is (Z1, Z2, Z3)-stable a new concept developed in this paper or does it already exist in the literature? If so, what is the citation for (Z1, Z2, Z3)-stable.
6) It is confusing to me that authors use the word 'stable' to talk about approximation quality in Sec 4. More specifically, how does (Z1, Z2, Z3)-stable relate to the general definition of stability in PDEs [L93-95]? An "equation is stable if the solution of the perturbed PDE converges to the exact solution as the perturbations approach zero [6]"


**Limitations:**

- There is no discussion of limitations nor of negative societal impacts.
- It would be very helpful if the limitation section could list and analyze the assumptions in Definitions/Theorems 4.1-4.4 and discuss what the extra steps would be to adapt the Definitions to other classes of PDEs beyond HJB.
- Are the theorems truly applicable to *all* variations of the HJB equations?
- What does (Z1,Z2,Z3)-stable mean and what does it *not* mean?
- It would be helpful if the authors mention that, while the work is very theoretical, it could be used to improve numerical modeling of applications that violate the NeurIPS ethical guidelines.


**Strengths And Weaknesses:**

Strength:
- The work is significant. The broader field of learning surrogate models of PDEs can have wide-reaching impacts in physics, chemistry, fluid dynamics, biology, climate modeling and more. Within the field of phyisics-informed machine learning, the authors have identified and addressed a very important research question: Does there exist theorems that connect properties of PDEs to the optimal choice of PINN loss function. The choice of HJB equations is important and developing the theory on HJB equations only should be sufficient for acceptance.
- The authors provide rigorous and sound theory to prove that a learned solution to HJB is (Z1, Z2, Z3)-stable iff the physics-informed loss uses Lp loss with sufficiently large p.
- The authors clearly state the research question [L34-35a] and include necessary background math, e.g., Def. 4.2.

Weaknesses:
- One of the authors main contributions seems to be the notion of (Z1, Z2, Z3)-stable. However, it is still unclear to me 1) if (Z1, Z2, Z3)-stable actually is a novel concept that the authors came up with, 2) what the concept can and cannot be used for, and 3) how the concept relates the 'stability' in PDEs (see Qestuions 5-6).
- It is challenging to fully evaluate originality of the work, because the related works section is very sparse. I have inluded some questions to evaluate originality and am willing to raise my score.
- The empirical evaluation does not fully support the theorems, nor does it fully answer the research question. I have included some questions to evaluate the quality of the empirical section.

---

> ### Author Response · Authors · 2022-08-01
> **Response to Reviewer hCTD (2/2)**
>
>
> **Response to Question 4.**
> There are a few works that use GAN loss in PINN training [7-9], mainly in a heuristic way. In contrast, we build a mathematical framework to study the relationship between PDE and PINN loss, which can guide the practitioners to choose loss functions (e.g., $L^\infty$) in a principled way.
> Another disadvantage of the GAN training is the well-known hyper-parameter sensitivity and optimization instability, while our algorithm doesn't have such issues.
>
> [7] Yang, Yibo, and Paris Perdikaris. "Adversarial uncertainty quantification in physics-informed neural networks." Journal of Computational Physics 394 (2019): 136-152.
>
> [8] Daw, Arka, M. Maruf, and Anuj Karpatne. "PID-GAN: A GAN Framework based on a Physics-informed Discriminator for Uncertainty Quantification with Physics." Proceedings of the 27th ACM SIGKDD Conference on Knowledge Discovery & Data Mining. 2021.
>
> [9] Bullwinkel, Blake, et al. "DEQGAN: Learning the Loss Function for PINNs with Generative Adversarial Networks." ICML 2022 2nd AI for Science Workshop.
>
>
> **Response to Question 5.**
> The concept $(Z_1, Z_2, Z_3)$-stable is first proposed in this paper, inspired by the stability theory in PDE literature.
>
> Stability is an important concept in analyzing PDEs, which have a much longer history than PINN. Roughly speaking, a PDE is stable if small perturbations in the equation can only lead to a slight change in its solution.
>
> For PINNs, we found the stability theory is essential in understanding the plausibility of the loss function. However, we need to customize the concepts to capture some unique characteristics of PINN, such as the PDE residual and boundary residual in the loss terms. Therefore, we propose the notion of $(Z_1, Z_2, Z_3)$-stable, which can help better characterize the asymptotic behavior of the learned models.
>
> **Response to Question 6.**
> Thanks for the question. The question helps us realize that there are some confusing parts in the paper when introducing the concept of stability. Here we make a more precise explanation and will revise the paper accordingly.
>
> Intuitively, for any function $u_{\theta}$ with a small loss, it can be imagined that there is another PDE, slightly different from the original PDE, whose solution is $u_{\theta}$. Here the term "slightly different" is defined by the loss. Then the central question becomes whether these two "slightly different" PDEs will always have slightly different solutions，which relates to the stability of the PDE. If the answer is yes (i.e., the PDE is stable), a small-loss solution will always approximate the true solution well.
>
> To be specific, suppose we obtain an approximate solution $u_\theta$ whose loss is small.
> Then $u_\theta$ corresponds to the solution to a perturbed PDE,
> $$
>     \begin{cases}
>     \mathcal{L}u(x)=\varphi(x)+R_1(x)& \quad x\in\Omega\\\\
>     \mathcal{B}u(x)=\psi(x)+R_2(x)& \quad x\in\partial\Omega,
>     \end{cases}
> $$
> where $R_1:=\mathcal{L}u_\theta -\varphi,\ R_2:=\mathcal{B}u_\theta-\psi$ can be seen as perturbations. Ideally, one would hope the approximate solution can be close to the exact solution when the perturbation is sufficiently small, and this is what "stable" refers to.
>
> **Response to limitations.**
> Thanks for the comments. These comments definitely figure out the potential of this research direction. We have responded to some of the points in the list of limitations above and will respond to other points here. Our definition of $(Z_1,Z_2,Z_3)$-stable is general and could be adapted directly to other PDEs. But determining whether certain PDE is stable and which loss is suitable to solve it may require problem-dependent analysis. This would be a promising future direction. HJB is a large class of PDEs that consists of instances with various properties. In the paper, we are careful not to make a general statement to cover all HJB equations. But we believe our technique can be used to analyze any given HJB class.
>
> May you have any further questions, please tell us and we are willing to address your concerns.

---

> ### Author Response · Authors · 2022-08-01
> **Response to Reviewer hCTD (1/2)**
>
> Thanks for your careful review! We respond to each question as below.
>
> **Response to Question 1.**
> Thanks for the suggestion! We further conduct experiments to compute $L^2$ and $W^{1,1}$ relative errors of our model and the baseline PINN method on the 100-dimensional HJB Equation (Eq.(12)). The results are shown below:
>
> |                           | $L^1$         | $L^2$         | $W^{1,1}$     |
> | ----                      | ----          | ----          | ----          |
> | Original PINN             | 3.47%      | 4.25%      |  11.31%    |
> | Adaptive time sampling    | 3.05%      | 3.67%      |  13.63%    |
> | Learning rate annealing   | 11.09%     | 11.82%     |  33.61%    |
> | Curriculum regularization | 3.40%      | 3.91%      |  9.53%     |
> | Ours                      | **0.27**%  | **0.33**%  |  **2.22**% |
>
> Clearly, our approach significantly outperforms the baselines by a large margin under all these evaluation metrics. We will include these results in our paper in the revision.
>
> **Response to Question 2.**
> In the paper, we are careful not to make a general statement to cover all high-dimensional PDE problems as the property of different PDEs can vary significantly. However, we believe some general mathematical tools should be developed to analyze Physics-Informed Loss rigorously. This work takes the first step to tackling the problem, and we will investigate more in the future.
>
> **Response to Question 3.**
> Thanks for your reference! Due to space limitations, we selected representative papers in the related work section. Han et al. (reference [11] in our paper) cast the problem into a backward stochastic differential equation (BSDE) which is further modeled by neural networks. Your mentioned work by Nüsken and Richter and many others [1-6] are all based on this framework. As is discussed in Line 83-87, this approach only learns the solution on a pre-defined time frame, while our PINN-based approach can learn the solution for any time frame. To the best of our knowledge, few prior works have applied PINN to learn any-frame solution for high dimensional HJB Equation, possibly due to the misuse of $L^2$ loss. We will add those references to the paper to make the related work section more comprehensive.
>
> [1] Pereira, Marcus, et al. "Learning deep stochastic optimal control policies using forward-backward sdes." Robotics: science and systems (2019).
>
> [2] Yu, Yajie, Bernhard Hientzsch, and Narayan Ganesan. "Backward deep BSDE methods and applications to nonlinear problems." arXiv preprint arXiv:2006.07635 (2020).
>
> [3]Pereira, Marcus, et al. "Feynman-kac neural network architectures for stochastic control using second-order fbsde theory." Learning for Dynamics and Control. PMLR, 2020.
>
> [4]Beck, Christian, et al. "Deep splitting method for parabolic PDEs." SIAM Journal on Scientific Computing 43.5 (2021): A3135-A3154.
>
> [5]Pham, Huyen, Xavier Warin, and Maximilien Germain. "Neural networks-based backward scheme for fully nonlinear PDEs." SN Partial Differential Equations and Applications 2.1 (2021): 1-24.
>
> [6] Davey, Ashley, and Harry Zheng. "Deep learning for constrained utility maximisation." Methodology and Computing in Applied Probability 24.2 (2022): 661-692.

---

### Author Response · Authors · 2022-08-01
**General Response**

We thank AC for handling this paper and thank all the reviewers for their kind help and useful suggestions. The comments have enlightened us to ponder how to improve the quality of our submission.

We will add some more discussions and new experimental results to the paper, including:

+ The performance comparisons between our model and baselines under more evaluation metrics, e.g., $L^2$ norm and $W^{1,1}$ norm, and more equations.
+ Detailed discussions on the failures of $L^p$ loss with large but finite $p$ that are shown in Section 6.2.
+ Error/loss-vs-time plots in the experiment part.
+ More discussions on the related work, theoretical framework, and limitations.

Due to the intensive discussions and conducting multiple experiments during the rebuttal, we can hardly finish the revision before the rebuttal deadline. We will update it as soon as possible.

Thanks!

Paper 1876 Authors

---

### Author Response · Authors · 2022-08-07
**General Response: Paper Revision**

We have finished the paper revision and uploaded the updated version of our paper. The amendments are:

+ We rewrite the derivation of equation, taking state-dependent control functions into consideration (Appendix B).
+ We add performance comparisons between our model and baselines under more evaluation metrics, including $L^2$ norm and $W^{1,1}$ norm (Appendix G.2).
+ We add comparisons with baselines on more equations (Appendix G.3).
+ We show the time series of training loss and test error of original PINN and our method (Appendix G.4).
+ We make detailed discussions on the failures of $L^p$ loss with large but finite $p$ (Appendix H).
+ We add discussions on limitations and future directions (Section 7).

The author-reviewer discussion deadline is approaching. We sincerely hope that the reviewers can re-evaluate the quality of our work based on our responses and revision. May you have any further questions, please feel free to discuss with us in recent days and we are willing to address your concerns.

Regards,

Paper 1876 Authors

---

### Meta-Review · Area_Chair_friS · 2022-09-04

**Recommendation:** Accept
**Confidence:** Less certain

**Metareview:**

The reviewers reached a consensus that this paper meets the bar for being accepted at NeuRIPS, and therefore the AC recommends acceptance. Please refers to the reviews and author's responses for reviewers' opinion on the strength and weakness of the paper.

**Award:**

No

---

### Decision · Program_Chairs · 2022-09-14

Accept